# Avoidable diet-related deaths and cost-of-illness with culturally optimized modifications in diet: The case of Brazil

**Eliseu Verly, Jr** [1]*, **Ísis Eloah Machado** [2], **Adriana Lúcia Meireles** [3], **Eduardo A. F. Nilson** [4]

**1** Department of Epidemiology, Institute of Social Medicine, Rio de Janeiro State University, Rio de Janeiro, RJ, Brazil, **2** Medicine School, Federal University of Ouro Preto, Ouro Preto, MG, Brazil, **3** School of Nutrition, Federal University of Ouro Preto, Ouro Preto, MG, Brazil, **4** Center for Epidemiological Research in Nutrition and Health, University of São Paulo, São Paulo, SP, Brazil

* eliseujunior@gmail.com

**Data Availability Statement:** The datasets analyzed during the current study are available on the Brazilian Institute of Geography and Statistics

## Abstract

### Background

Dietary risk factors have an important impact on premature deaths and disabilities due to non-communicable diseases. In this study, we perform diet optimization to design different dietary scenarios taking into account food prices and preferences and evaluate the number of deaths that would be prevented as well as the economic burden and costs from the health system that would be saved in Brazil.

### Methods

We used dietary intake and food prices data from the nationwide Household Budget Survey (HBS) and the National Dietary Survey (NDS) 2017–2018. Linear programming models were performed to design five scenarios which different sets of key diet modifications at the least deviation from the baseline consumption. Comparative risk assessment models were used to estimate the health impacts of optimized dietary changes on mortality and the economic impacts on morbidity (hospitalizations) and premature deaths.

### Results

The optimized diets were, on average, more expensive than the baseline diets, varying from Int\$ (international dollar) 0.02/day to 0.52/day/adult. The number of deaths prevented or postponed varied from 12,750 (10,178–15,225) to 57,341 (48,573–66,298) according to the different scenarios. The diet modifications would save from 50 to 219 million in hospitalizations and from 239 to 804 million yearly in productivity losses with the reduction of premature deaths.

### Conclusion

A substantial number of deaths and costs due to hospitalization and productivity losses would be avoidable even with small changes in diets. However, even the cheapest

(IBGE) website, [https://www.ibge.gov.br/en/
statistics/social/population/25610-pof-2017-2018-
pof-en.html?=&t=o-que-e]. Variable names,
description, and contents are in Portuguese. The
Excel file used for running the PRIME is available
on https://www.euro.who.int/en/health-topics/
disease-prevention/tobacco/publications/2019/
ncdprime-modelling-the-impact-of-national-
policies-on-noncommunicable-disease-ncd-
mortality-using-prime-a-policy-scenario-
modelling-tool-2019. SAS codes used for diet
optimization are available from the corresponding
author on reasonable request.

**Funding:** EVJ was funded by FUNDAÇÃO CARLOS
CHAGAS FILHO DE AMPARO À PESQUISA DO
ESTADO DO RIO DE JANEIRO (FAPERJ - www.
faperj.br), grant number E26/201.332/2021. IEM
was funded by CONSELHO NACIONAL DE
DESENVOLVIMENTO CIENTÍFICO E
TECNOLÓGICO (CNPq - www.cnpq.gov.br) and
Coordenação Geral de Alimentação e Nutrição do
Departamento de Promoção da Saúde da
Secretaria de Atenção Primária à Saúde do
Ministério da Saúde (CGAN/DEPROS/SAPS/MS),
grant number 442636/2019-9. The funders had no
role in study design, data collection and analysis,
decision to publish, or preparation of the
manuscript.

**Competing interests:** The authors have declared
that no competing interests exist.

intervention might be prohibitive for deprived families, yet subsidies and social policies could contribute to improving diets.

## Introduction

Noncommunicable chronic diseases (NCD) are the largest cause of morbidities and mortality worldwide. The last estimate from the Global Burden of Disease Study (2019) estimated that about 74.4% of the deaths globally were due to NCD [1]. Among the main risk factors for NCD, dietary risks stand as one of the most relevant. About 18.9% of all deaths from NCD in the world could be attributable to dietary risks. The fraction of deaths attributable to dietary risks varies according to the country. According to the GBD, in 2019, 75.9% of all deaths in Brazil were due to NCD and 14.1% of the NCD deaths were attributable to dietary risks, from which the leading risks were a diet high in red meat, a diet low in whole grains, a diet high in sodium, and a diet low in vegetables [2]. The population attributable fraction concept is very useful for public health and policies once it can be used to estimate the impact of a risk factor, in terms of mortality, years of life lost (YLLs), years lived with disability (YLDs), disability-adjusted life-years (DALYs), and then provide with a more objective metric to measure the potential impact of interventions. In other words, it represents the number of deaths that would be prevented if the exposure were removed in the past.

In addition, NCD represents major costs to the health systems and economies around the world. In Brazil, the direct and indirect costs of cardiovascular diseases increased 17% from 2010 to 2015, reaching US$ 9.6 billion in 2015 [3]. Also, the attributable costs to excessive sodium intake in Brazil, including hospitalizations, outpatient care, medications, and the burden of premature deaths, totaled US$ 945 million in 2017 [4]. The yearly costs of diseases attributable to obesity to the Brazilian National Health System increased from US$ 269.6 million to US$ 636.7 million, from 2011 to 2018 [5]. Thus, key diet modifications are effective strategies to reduce the burden of disease and consequently the cost spent by the health system. However, diet modifications may not be feasible unless they are affordable and culturally acceptable [6,7].

Many studies have assessed the relationship between diet cost and quality. In general, healthier diets are more expensive compared to nutritionally poor diets [8]. Also, theoretical studies with linear programming have demonstrated that, at least for part of the population, reaching nutritional goals necessarily increases the cost [9]. In countries with marked social inequality, such as Brazil, improving diet quality might be prohibitive for many families, particularly in the last years, where the percentage of families living with some degree of food insecurity increased from 22.9% in 2013 to 55.2% in 2022 (4.2% and 9% for severe food insecurity, respectively) [10]. Although traditional diets based on staple foods can be, on average, cheaper than the current diet [9], the adequacy of key items such as fats, sodium, and potassium increases the cost. In low-income families, this increase was estimated to be up to 40% in relation to the current cost. Thus, identifying feasible modifications that are effective in terms of preventing deaths and cost provides important insights for public health and policies.

Several studies have performed diet optimization to design diets to attain nutritional goals while respecting other aspects of the diets, such as food preferences and cost constraints. It has also been used to verify the lowest possible cost when moving from the current to optimum diets taking into account the variability in food prices and preferences across the population [11–13]. In general, the nutritional targets of these studies are nutrient intakes or food group recommendations, such as those to prevent NCD. Although informative, it is not

straightforward to translate these optimized diets into metrics that could be objectively used to define targets of interventions and policies, such as the number of avoidable deaths, since it depends on the assessment of the risk for each subpopulation at different levels of dietary exposure. In this study, we aim (i) to perform diet optimization to design different dietary scenarios taking into account food prices and preferences; (ii) to perform the comparative risk assessment approach to estimate the number of deaths that would be prevented associated with the output optimized diets; and (iii) to estimate the economic burden and costs from the health system that would be saved associated with those diets.

## Methods

### Data sources

We used data from the nationwide Household Budget Survey (HBS) 2017–2018, which collected information on household food purchases, and the National Dietary Survey (NDS) 2017–2018, which collected information on individual food consumption. NDS was simultaneously performed in a random subsample of ~35% of the HBS, thus food consumption and purchase were collected in the same household and time frame. Both surveys were conducted by the Brazilian Institute for Geography and Statistics. The samples included 57,920 households (HBS) and 20,112 households (NDS). Data collection in each stratum was uniformly distributed throughout the four trimesters to account for seasonal variations in both food intakes and prices. More information on the surveys and data collection can be found elsewhere [14,15].

### Baseline dietary intakes

Two non-consecutive 24h recalls based on the Automated Multiple-Pass approach [16] were answered by all household members aged 25 years or older (n = 37,687). Detailed information on portion size, amount consumed, cooking method, and time and place of consumption were provided for each food reported. A total of 1591 food items were reported in the NDS (coffee, tea, and alcoholic beverages were not considered in this study). From this list, the items were aggregated if they were different types, different cooking methods, or different meat cuts of the same food (e.g., different types of orange into "orange", different beef cuts into "beef", etc.). The final list comprised 83 to 85 foods, depending on the age-sex group. The food list, mean food intakes, and the number of foods considered in each age-sex group are shown in (**S1 and S2 Tables in S1 File**).

   Mean food intakes were obtained for each age-sex group and used as starting points to design optimized diets using linear programming models. Twelve age-sex subgroups were defined: 25-30y, 31-39y, 41-49y, 51-59y, 61–69, and $>= 70$, all age groups stratified by sex. The Brazilian Food Composition database [17] was used to obtain nutrient content in both observed and optimized diets. Nutrient composition of foods clustered from food subtypes (e.g. different types of rice into 'rice') was obtained as the mean composition of the food subtypes weighted by their frequency of reporting in the NDS.

### Food prices

Food prices were extracted from the HBS database, where each household registered the amount and price of each food product purchased, further converted into prices per 100g of edible portion. Considering the variation in food prices throughout the collection (12 months), all prices were deflated to the same reference date (January 31st 2018) using official inflation rates (National Consumer Price Index–INPC). Prices of food items were obtained as the mean

price over the food subtypes (e.g., different types of oranges into 'orange', or different types of fish and seafood into 'fish and seafood') weighted by their frequency of reporting in the budget survey, i. e., expensive, but low frequently eaten food subtype has minor importance to the food item price. Food prices were matched to the corresponding food item declared as consumed in the NDS.

## Diet modeling

Linear programming models were performed to design healthier diets that most resemble the current diets. A model for each age-sex group was performed for each intervention scenario. The decision variables were the foods reported by each age-sex group as described above. Although modifications in food quantities are necessary to move the current to healthier diets, the sum of the difference between each optimized food quantity and observed food intake should be as least as possible. The objective function in the LP models was defined to minimize the sum of the absolute difference between each optimized and baseline food quantity, described as follows:

$$Minimize\ Y = \sum_{i=1}^{i=g} |\frac{Q_i^{opt} - Q_i^{obs}}{Q_i^{obs}}|$$

Where Y represents the objective function to be minimized, $Q_i^{opt}$ is the quantity of the food item $i$ in the optimized diet, $g$ is the total number of food items, $Q_i^{obs}$ is the mean quantity of $i$ in the observed diet. Once it is a nonlinear function, a linearization to include a set of linear constraints was performed following the procedure described in [18]. The decision variables were the foods that composed the reported diets in each age-sex group. Several types of constraints were introduced into the models as described below.

**Dietary constraints (Dietary scenarios).** We designed five scenarios in which a set of key healthy dietary components were progressively increased (beneficial for health) and decreased (detrimental for health). We modeled our dietary goals using the Preventable Risk Integrated Model (PRIME). PRIME includes diet-outcome pairs from published meta-analyses of epidemiological studies: prospective cohort studies for NCD mortality and randomized controlled trials for blood cholesterol and blood pressure. In this study, we focused on dietary risk factors only. The risk factors, outcomes, and relative risks used here are shown in Table 1.

Table 1. Dietary risk factors, outcomes, unit of change and effect size of PRIME model.

| Dietary risk factor | Outcome | Unit of change | RR[c] (95% CI) |
|---|---|---|---|
| Fruit | CHD[a] | 106 g/day increase | 0.93 (0.89; 0.96) |
| | Stroke | 106 g/day increase | 0.89 (0.85; 0.93) |
| | Lung cancer | 80 g/day increase | 0.94 (0.90; 0.97) |
| Vegetables | CHD[a] | 106 g/day increase | 0.89 (0.83; 0.95) |
| Fiber | CHD[a] | 10g/day increase | 0.81 (0.72; 0.92) |
| | Stroke | 7 g/day increase | 0.93 (0.88; 0.98) |
| | Colorectal cancer | 10 g/day increase | men: 0.88 (0.78; 0.99) |
| | | | women: 0.92 (0.87; 0.98) |
| Saturated fat | total serum cholesterol (nmol/L) | 1% of total calories increase | 0.052 (0.045; 0.058)[d] |
| PUFA[b] | total serum cholesterol (nmol/L) | 1% of total calories increase | -0.026 (-0.034; -0.018)[d] |
| Dietary cholesterol | total serum cholesterol (mg/L) | 1 mg/d increase | 0.001 (0.001, 0.001)[d] |
| Salt | Systolic blood pressure (mmHg) | 6 g/day reduction | -5.80 (-2.50; -9.20)[d] |

Source: Scarborough et al [19].

[a]CHD: Coronary Heart Disease; [b]PUFA–Poly-unsaturated fat acids; [c]RR–Relative Risk; [d] Absolute change in the outcome level (linear regression coefficient).

**Table 2. Dietary constraints used in the five optimized scenarios [a].**

| Dietary item | Sc#1 | Sc#2 | Sc#3 | Sc#4 | Sc#5 |
|---|---|---|---|---|---|
| Fruit (g) | +25 | +50 | +75 | +75 | +75 |
| Vegetables (g) | +25 | +50 | +75 | +75 | +75 |
| Fiber (g) | +2 | +4 | +6 | +6 | +6 |
| PUFA[c] (%)[b] | - | - | - | +20 | +20 |
| Saturated fat (%)[b] | - | - | - | -20 | -20 |
| Cholesterol (%)[b] | - | - | - | -10 | -30 |
| Sodium (%)[b] | - | - | - | - | -10 |

[a] scenarios represent the amount to change in a daily basis.

[b] percentage in relation to the baseline.

[c] PUFA: Poly-unsaturated fatty acids.

Dietary constraints were introduced according to the scenario as described in **Table 2**. These constraints indicate how much each dietary item in the optimized diets should move in relation to the baseline diets, for each age-sex group. For example, the mean baseline fruit intake in 25-30y-male was 151g, then in Sc#1, the optimized diet for this age-sex group should achieve 151g+25g (176g). For poly-unsaturated fatty acid (PUFA), saturated fat, cholesterol, and sodium, the changes were described as percentages in relation to the baseline intakes.

In all models, total energy content was constrained to be equal to the mean baseline energy intake estimated in each age-sex group.

**Food acceptability constraints.** Boundaries limiting changes in food quantities were introduced in the models to make sure that optimized food quantities are not beyond the range of intakes observed in the population. The boundaries were defined as the first and the 99th percentiles of intake of each individual food item stratified by age-sex groups. Constraints for food groups were also introduced in the models, which was defined as the 99th percentile of intake of 24 food groups, also stratified by age-sex group. Age-sex-specific food constraints are presented in **S1 and S3 Tables in S1 File**. The age-sex-specific optimized food items and food group quantities should not be lower or higher than the corresponding boundaries. However, the upper boundaries for some food groups are lower than the dietary goals, which is the case of FV. In this case, the upper boundary for FV was progressively extended (by every 25%) until the model finds a feasible solution compatible with the dietary goals. Linear programming models were performed using the Optmodel Procedure from the software SAS OnDemand.

## Attributable deaths

The attributable deaths in each policy scenario were estimated using the Preventable Risk Integrated ModEl (PRIME), a comparative risk assessment methodology suitable to estimate the health impact of changes in the behavioral risk factors for non-communicable diseases on population and age-sex specific NCD mortality [19]. For the dietary risk factors, the PRIME model considers the impacts through associations mediating factors such as body mass index (BMI), blood pressure, and blood cholesterol, which are parametrized based on published meta-analyses of epidemiological studies. In the present study, data on mortality from NCDs and population demographics were obtained, respectively, from publicly available tables of the Brazilian Mortality Information System (SIM) for 2019 and the Brazilian Institute of Geography and Statistics (IBGE) for 2017. Demographic and mortality data were stratified by gender and 5-year age bands, and mortality data were based on the World Health Organization (WHO) International Classification of Diseases 10 (ICD 10).

The counterfactual risk factor distributions were based on the 5 scenarios (**Table 2**). The mean intakes of each scenario were modeled with linear programming, and the coefficient of variation was assumed to be the same as of the baseline intakes. The counterfactual distributions were defined for each age-sex group. The PRIME first estimates the Potential Impact Fraction (PIF) which stands for the percentage of each cause-specific death due to a change in a given risk factor. PIFs were calculated for each outcome-risk pair and age-sex group using this equation:

$$PIF = \frac{\int_0^m RR(x)P(x)dx - \int_0^m RR(x)P'(x)dx}{\int_0^m RR(x)P(x)dx}$$

where RR(x) is the relative risk of the outcome $o$ at the exposure level $x$, P($x$) is the baseline population at exposure level $x$, and P'($x$) is the counterfactual population at exposure level $x$. The PIFs for different risk factors were combined multiplicatively using the equation:

$$PIF_{total} = 1 - \prod_{i=1}^{n}(1 - PIF_i)$$

## Attributable cost

To estimate the direct and indirect costs related to dietary risks (salt, fats, fruits, vegetables, and fibers), we applied the corresponding age-sex-specific PIFs related to each health outcome for each counterfactual scenario as estimated previously using the PRIME. Then, we estimated the proportion of direct costs to the Brazilian National Health System (hospitalizations) and indirect costs related to premature deaths in each diet scenario, expanded from a costing tool developed for hypertension and related cardiovascular disease [20].

The costs of hospitalizations were based on a top-down approach through the identification of the direct costs of diet-related NCDs by specific International Classification of Disease (ICD) codes and valuations, based on administrative data obtained from the Hospital Information System (SIH/SUS) [21], which registers all hospitalizations provided by the Brazilian National Health System (SUS). In 2019, hospitalizations provided by the SUS corresponded to about 70% of total hospitalizations in Brazil (estimated by the Observatory of Hospital Policy and Management using data from the Hospital Information System and National Regulatory Agency for Private Health Insurance and Plans, available in Portuguese on https://observatoriohospitalar.fiocruz.br/).

The costs of premature death were based on the estimated Years of Productive Life Lost (YPLL), using the Human Capital Approach [22] to calculate the present value of potential time in the workforce (the measure of productivity) using country-specific data for 2019, such as the pension age (60 years for women, and 65 years for men), the average national wage and the labor force participation estimates [23], considering a discount rate of 2% per year [20].

All costs were converted into international dollars (Int$), based on the purchasing power parity (PPP) of Brazil for the year 2019 (1 Int$ equals 2.28 Brazilian Reals on Jan 31[st] 2019) [24].

## Uncertainty analysis

Uncertainty intervals are calculated based on 5,000 Monte Carlo iterations which allow the relative risks to randomly vary according to the distribution described in the literature and the intake distribution of dietary inputs to randomly vary according to the sample standard error.

## Results

### Model feasibility

In general, no solution for any scenario was reached using the food acceptability and food group constraints derived from the population intake distribution. Thus, progressive flexibility

in FV and nuts constraints were necessary. Cholesterol constraint was not attained in 70y
+ Females; sodium constraint was not attained in 50-60y Females; PUFA constraints were not
attained in 30-40y Males and 60y+ Females. In all these cases, the constraints were set to be
equal to or higher than the baseline mean intakes.

## Optimized food and nutrient quantities and diet cost

Table 3 compares food and food group mean baseline intakes with the five optimized scenar-
ios food and food group content. In the scenarios only constrained by FV and fiber, the
increase in these components led to an increased cost up to Int$ 0.33/day (increasing 150g FV/
day), and was compensated mainly by a reduction in red and processed meats (from 99.8 to
84.9g at this FV increase). The other food quantities remained stable over scenarios #1, #2, and
#3. In scenario #4, where fats (saturated and poly-unsaturated) and cholesterol were also con-
strained, the cost increase was slightly reduced compared with scenario #3 (from Int$ 0.33 to
Int$ 0.24/day). This scenario was marked by a higher increase in FV (+215g/day), beans
(+36g/day), nuts (+3.2g/day), and a reduction in cookies, cakes, dairy, bread, snacks, sweets,
and SSB. In scenario #5, the introduction of the sodium constraint in the models led to a

**Table 3. Food quantities (in grams/day) and diet cost in the baseline and five optimized scenarios.** Brazil, 2019.

|  | Baseline | s.e. | Sc#1 | Sc#2 | Sc#3 | Sc#4 | Sc#5 |
|---|---|---|---|---|---|---|---|
| Total cost (Int$)[a] | 3.39 | 0.16 | 3.41 | 3.51 | 3.72 | 3.63 | 3.91 |
| FV[b] | 166.7 | 3.30 | 216.9 | 266.7 | 324.7 | 381.7 | 510.2 |
| Fruits | 77.68 | 1.90 | 102.7 | 127.7 | 152.7 | 155.9 | 243.3 |
| Leafy vegetables | 12.20 | 0.50 | 17.20 | 24.17 | 32.80 | 27.16 | 54.92 |
| Other vegetables | 64.63 | 1.50 | 79.77 | 90.69 | 106.4 | 171.5 | 157.0 |
| Tuber | 35.60 | 0.80 | 40.72 | 45.60 | 50.60 | 62.17 | 30.79 |
| Whole cereals | 10.45 | 1.63 | 15.27 | 16.31 | 25.45 | 23.86 | 20.36 |
| Rice | 145.2 | 10.40 | 150.2 | 150.2 | 150.2 | 157.6 | 158.2 |
| Pasta | 38.88 | 2.60 | 39.06 | 43.88 | 41.38 | 48.24 | 24.50 |
| Beans | 166.3 | 13.20 | 172.6 | 176.3 | 178.2 | 190.9 | 198.1 |
| Nuts | 0.59 | 0.00 | 0.59 | 1.54 | 3.54 | 3.79 | 3.86 |
| Dairy | 94.79 | 2.90 | 92.82 | 90.65 | 85.72 | 80.51 | 80.51 |
| Olive oil | 2.63 | 0.10 | 2.63 | 0.27 | 0.12 | 0.12 | 2.03 |
| Fish and seafood | 23.91 | 1.40 | 23.91 | 21.58 | 18.92 | 36.73 | 28.94 |
| Red meat | 83.05 | 7.10 | 75.74 | 73.05 | 73.05 | 58.69 | 53.53 |
| Processed meat | 16.75 | 1.43 | 12.48 | 11.90 | 11.90 | 6.63 | 5.44 |
| Poultry | 52.21 | 2.30 | 47.21 | 47.21 | 47.21 | 49.73 | 54.45 |
| Eggs | 16.36 | 1.00 | 16.36 | 11.52 | 11.52 | 7.05 | 7.05 |
| Butter | 1.00 | 0.00 | 1.00 | 1.00 | 1.00 | 0.70 | 0.70 |
| Margarine | 9.52 | 0.30 | 10.31 | 8.93 | 7.62 | 13.58 | 13.83 |
| Cookies | 13.40 | 1.00 | 13.30 | 13.30 | 13.20 | 10.20 | 10.60 |
| Cakes | 12.29 | 0.40 | 12.29 | 13.77 | 7.72 | 5.87 | 5.64 |
| Bread | 51.81 | 2.80 | 54.85 | 56.53 | 56.02 | 39.45 | 36.78 |
| Sweets | 8.47 | 0.40 | 8.47 | 8.47 | 3.47 | 1.30 | 1.27 |
| Snacks | 4.26 | 1.90 | 4.26 | 4.26 | 3.95 | 4.13 | 3.84 |
| SSB[c] | 58.23 | 9.10 | 56.68 | 54.18 | 53.23 | 54.59 | 58.23 |

[a] Int$: International dollar (1 Int$ equals 2.28 Brazilian Reals in Jan 31st 2019).

[b] FV: Fruit and vegetables.

[c] SSB: Sugar-sweetened beverages.

**Table 4. Mean nutrient intakes and nutrient contents (per day) in the baseline and five optimized scenarios.** Brazil, 2019.

| | Baseline | s.e. | Sc#1 | Sc#2 | Sc#3 | Sc#4 | Sc#5 |
|---|---|---|---|---|---|---|---|
| Energy (kcal) | 1724 | 77 | 1724 | 1724 | 1724 | 1724 | 1724 |
| Carbohydrates (%Kcal) | 54.8 | 50.0 | 56.8 | 59.3 | 59.6 | 60.2 | 60.3 |
| Protein (%Kcal) | 19.2 | 0.8 | 18.4 | 18.2 | 18.2 | 17.4 | 17.0 |
| Total fats (%Kcal) | 28.8 | 1.6 | 27.9 | 25.9 | 25.8 | 26.3 | 26.9 |
| Saturated Fat (%Kcal) | 9.3 | 1.0 | 8.8 | 8.1 | 8.1 | 7.5 | 7.5 |
| MUFA[a] (%Kcal) | 9.5 | 1.0 | 9.1 | 8.0 | 7.8 | 7.6 | 8.2 |
| PUFA[b] (%Kcal) | 7.0 | 1.0 | 7.0 | 7.0 | 7.0 | 8.4 | 8.4 |
| trans-fats (%Kcal) | 0.6 | 0.1 | 0.6 | 0.6 | 0.5 | 0.5 | 0.5 |
| Cholesterol (mg) | 271.9 | 14.9 | 254.1 | 222.8 | 219.3 | 201.6 | 186.5 |
| Fiber (g) | 22.8 | 1.2 | 24.8 | 26.8 | 28.8 | 30.7 | 33.4 |
| Sodium (mg) | 2295.1 | 132.9 | 2273.2 | 2291.8 | 2279.0 | 2295.1 | 2065.6 |

[a]MUFA: Mono-unsaturated fatty acids.

[b] PUFA: Poly-unsaturated fatty acids.

higher cost increase (Int$ 0,52 compared with the baseline cost), and a more drastic increase in FV (+344g/day compared with the baseline consumption) while the remaining foods kept relatively constant in comparison with the previous scenario. The mean energy, carbohydrates, fats, fiber, cholesterol, and sodium contents in the baseline and in each optimized diet are shown in Table 4.

The estimated deaths preventable or postponable in each optimized diet scenario are detailed in Table 5, ranging from 12,750 (Scenario 1) to 57,341 deaths (Scenario 5), in 2019. Most deaths preventable or postponable in all scenarios are related to the increase in the consumption of fruits and vegetables and fibers, followed by changes in saturated fat, PUFA, cholesterol, and salt. Age-sex-specific, dietary risk-factor-specific, and cause-specific numbers of deaths preventable or postponable according to the intervention scenario are presented in **S1, S4, S5 and S6 Tables in S1 File**.

The changes in the optimized diets are reflected in the economic impacts of morbidity (hospitalizations) and premature deaths. By gradually increasing the nutritional profile of the diets, the direct cost savings in hospitalizations from diet-related NCDs almost quadrupled (Table 6) and the savings related to the indirect costs of premature deaths increased by 3.3 times (Table 7) from Scenario 1 to Scenario 5. Total savings considering direct and indirect costs of disease with the optimized diets ranged from approximately Int$ 289 million to Int$ 1.023 billion, in 2019. Cause-specific and dietary risk-factor-specific cost save according to the intervention scenario are presented in **S1 and S7 Tables in S1 File**.

**Table 5. Preventable or postponable deaths (uncertainty intervals) in the five optimized scenarios.** Brazil, 2019.

| | Sc#1 | Sc#2 | Sc#3 | Sc#4 | Sc#5 |
|---|---|---|---|---|---|
| Men | 5,858 | 11,746 | 16,736 | 17,953 | 30,351 |
| | (4,707; 6,973) | (9,578; 13,826) | (13,652; 19,759) | (14,668; 21,053) | (25,745; 35,035) |
| Women | 6,902 | 12,098 | 16,014 | 19,373 | 27,001 |
| | (5,202; 8,447) | (9,432; 14,568) | (12,632; 19,026) | (15,568; 23,066) | (22,674; 31,405) |
| Total | 12,750 | 23,830 | 32,739 | 37,317 | 57,341 |
| | (10,178; 15,225) | (19,241; 28,200) | (26,354; 38,589) | (30,468; 43,987) | (48,573; 66,298) |

**Table 6. Estimated saving costs (uncertainty intervals) in hospitalizations by the National Health System (Int$ million) in the five optimized scenarios.** Brazil, 2019.

|  | Sc#1 | Sc#2 | Sc#3 | Sc#4 | Sc#5 |
|---|---|---|---|---|---|
| Men | 25 | 54 | 74 | 79 | 129 |
|  | (20.3; 30.1) | (44.2; 62.7) | (60.3; 86.6) | (64.9; 92.1) | (107.7; 147.5) |
| Women | 25 | 44 | 56 | 68 | 91 |
|  | (17.9; 30.2) | (34.4; 53) | (44.6; 66.6) | (54.9–78.9) | (76; 103.8) |
| Total | 50 | 59 | 130 | 146 | 219 |
|  | (40.1; 58.9) | (78.6; 115.7) | (104.9; 153.3) | (121.2–171.0) | (183.7; 251.2) |

**Table 7. Estimated saving costs (uncertainty intervals) of productivity losses with the reduction of premature deaths (Int$ million) in the five optimized scenarios.** Brazil, 2019.

|  | Sc#1 | Sc#2 | Sc#3 | Sc#4 | Sc#5 |
|---|---|---|---|---|---|
| Men | 167 | 265 | 375 | 465 | 565 |
|  | (129.8; 202.6) | (211.5; 316.6) | (302.3; 444.4) | (379.7; 548.1) | (470.8; 649.9) |
| Women | 72 | 57 | 94 | 130 | 240 |
|  | (56.1; 87.6) | (45.3; 67.9) | (76; 111.7) | (106.1; 153.1) | (199.8; 275.9) |
| Total | 239 | 322 | 470 | 595 | 804 |
|  | (185.9; 290.2) | (256.8; 384.5) | (378.3; 556.1) | (485.8; 701.2) | (670.6; 925.7) |

With scenario 1, about 1,450 deaths [12,750/(0.023*365)] would be avoided and Int$ 33 million [(50+239)/(0.023*365)] would be saved for each 1 Int$/day increase in diet over a year; in scenario 5, these values would be 308 deaths and Int$ 5.4 million.

## Discussion

In this study, we estimated the cost of dietary intervention in different scenarios and how it would impact the number of diet-related deaths and health assistance spending in Brazil. In all cases, the optimized diets were, on average, more expensive than the baseline diets, varying from Int$ 0.02/day (BRL 0.45) to Int$ 0.52/day (BRL 1.18) per adult according to the nutritional targets introduced in each scenario. Number of deaths avoided also varied according to the scenario, from 12,750 (10,178; 15,225) to 57,341 (48,573; 66,298), which represent 1.03% and 4.63% of the total deaths registered in 2019 for the causes considered in this study. The reduction in deaths preventable by diet modifications would impact the health spending from the public health service, saving from 289 to 1,023 billion a year.

These results represent the cost for the lowest deviation from the observed to the optimized scenarios. We opted to focus on the food preference constraints instead of cost constraints. Both dimensions have been described as having an important role in food choices [25]. However, once they are measured on different scales, it is difficult to distinguish which one would play the highest influence on the individuals' choice. We assume that, while people's decisions take into account the price, maybe mostly in low-income households, people also do not buy what they are not willing to eat, even if they can afford it. It is in line with the fact that healthy foods such as fruit and vegetables in Brazil and most of the world are, in general, low regardless of the family income level in middle- and high-income countries. Thus, the actual dietary intakes (food quantities in the current diets) stand, at the most, for the balance between price and preferences. In the context of these analyses, it means that the overall optimized diets should be as close as possible to the baseline diets; in other words, people would be able to meet the dietary targets by keeping as close as possible to their current food habits. Scenarios with no cost increment or even at a lower cost compared with the baseline cost would be

feasible, however, they would demand higher diet modifications [11,18], which might not be acceptable or realistic.

It must be clear, however, which intervention would be more cost-effective, once cost increment compared with the baseline cost was observed in all scenarios. This cost increase may not be feasible for families in low socioeconomic status; thus, it should take place via price reduction of key foods (for example, subsiding fruit and vegetables). The cost for the government would be offset by the reduction in the spending on health assistance and the productivity gains due to the reduction in deaths and diseases. All scenarios have shown to be somewhat cost-effective; in the first two scenarios, increasing only FV and fiber, the cost increments were the lowest as well as the number of deaths prevented and the cost savings by the health system, when compared with the other scenarios. Besides the cost-effectiveness in scenarios 4 and 5 being lower, they also rely on the assumption that, in addition to the higher cost increment, people would need to tolerate more changes in their current diet.

It is worth noting that the epidemiological and economic dimensions of this study are subject to important time lag. The potential impact fraction estimated stands for a proportion of deaths that would be avoided if the exposure had been modified in the past, but we cannot predict how much would have cost the diet in each of the scenarios if they were adopted several years ago for two reasons. First, our optimized diets were designed to resemble the current diets as much as possible, which is likely to be somewhat different from diets in the past [26]. Second, price variation is determined, among other variables, by the demand and supply over the years. FV supply, for example, should have followed the increased demand for the quantities in the modeled scenarios to not increase the prices. The implication of these results in the future follows the same rationale. We don't know, indeed, how much will be the diet cost and the spending saved in the future once prices might vary for many reasons.

There are several studies assessing the impact of simulated dietary interventions. However, according to a review performed by Grieger et al [27], most were conducted with North American and European data, and most interventions were based on changes in nutrients or foods consumed in excess. Similarly, in Brazil, the studies on simulated interventions are mostly focused on critical nutrients, particularly on sodium [4,20,28], or on health outcomes, such as obesity, cancer, diabetes, and cardiovascular disease.

Few studies have assessed intervention with several dietary items at once, and usually, they do not account for the impact of food substitution when changing the target items. Food substitution modeling allows us to estimate the effect of replacing foods (healthy with unhealthy and vice-versa) that may impact nutrient intakes. According to Grieger et al [27], this is a clear gap in the studies included in the review once healthy foods are expected to have a different nutrient profile when compared to unhealthy ones. In our analysis, the modeled higher intake of FV led to a reduction in saturated fat content (scenario 3). Conversely, changing PUFA, saturated fat, and sodium content in diets led to an increase in FV (scenarios 4 and 5). All of these changes were considered when estimating the cost and the preventable deaths. In addition, the comparison with other studies is not straightforward once there are important variations in the assumptions, inputs, and interventions. The outcomes considered for each dietary risk, the systematic reviews or individual studies considered to obtain the relative risks, and the intervention set affect the population attributable fraction and consequently the number of deaths and the economic burden. As an example, Krueger et al [29] estimated the economic benefits of attaining FV recommendation in Canada. Unlike our analysis, they focused on FV for which the counterfactual scenario was the official FV consumption by everyone. As expected, the economic burden (US$ 4.39 billion) was much higher than we found in our most optimistic scenario (150g increase of FV).

## Strengths and limitations

Our study innovates by estimating the potential cost of the interventions for the households. It is critical to know to which degree families would afford healthy choices, besides providing information for supporting taxation, subsiding, and any other related policies.

One common limitation when modeling the mean intakes is that the distribution is assumed to be the same as before the intervention. In a real scenario, an increase in mean intake would be more feasible if the intake increased more among low consumers, compared to the high consumers at the baseline, which would reduce the variance of the intake distribution in a group. It would ultimately increase the population attributable fraction (PAF) once fewer people will be at the highest risk (low consumption) and fewer people will be over the theoretical maximum risk exposure level (high consumers).

Another important limitation is the underreporting, commonly observed in self-reported instruments to collect dietary intakes. A pooled analysis of several validation studies with biomarkers found a mean energy underreporting of 18% [30]. However, we do not know how much is the underreporting over age-sex groups. It is apparent that there is a substantial underreporting of energy intake in older groups, where the reported energy intakes were 1418 kcal and 1374 kcal for males and females 70y and older, respectively. It makes it more difficult to change the food quantities in such a restricted caloric set, that's why we relaxed the caloric content in some models. Underreporting in dietary intake also leads to underestimation in the diet cost. While the cost estimate is not accurate, the cost difference between the baseline and the interventions is still informative once they are subject to the same effect of the measurement error in dietary intakes (the scenarios are isocaloric).

Comparative risk assessment models such as PRIME for simulating the attributable deaths are adaptable to different settings and require inputs available in multiple settings, such as population-level estimates of risk factor distributions and disease-specific mortalities. Nevertheless, the estimated deaths delayed or averted do not consider the lifetime exposure to risk factors nor the time-lag effect between exposure and disease outcome. Therefore, the estimates in this study are not intended to predict the future but rather to estimate the difference between two or more policy or epidemiological scenarios.

Regarding the attributable direct and indirect costs of disease, the estimates are conservative because they only consider hospitalizations in the public health system and the costs of premature deaths. For example, other direct costs such as primary health care, outpatient consultations and procedures, and private or out-of-the-pocket payments, and the indirect costs of the study do not include the effects of presenteeism, absenteeism, sick leaves, and early retirements.

We concluded that a substantial number of deaths and costs due to hospitalization and productivity losses would be avoidable even with small changes in diets. However, even the cheapest intervention might be prohibitive for deprived families, yet subsidies and social policies could contribute to improving diets.

## Supporting information

**S1 File. S1–S7 Tables.**
(PDF)

## Author Contributions

**Conceptualization:** Eliseu Verly, Jr, Eduardo A. F. Nilson.

**Formal analysis:** Eliseu Verly, Jr, Eduardo A. F. Nilson.

**Funding acquisition:** Eliseu Verly, Jr, Ísis Eloah Machado, Adriana Lúcia Meireles.

**Methodology:** Eliseu Verly, Jr, Ísis Eloah Machado, Eduardo A. F. Nilson.

**Writing – original draft:** Eliseu Verly, Jr, Ísis Eloah Machado, Adriana Lúcia Meireles, Eduardo A. F. Nilson.

**Writing – review & editing:** Ísis Eloah Machado, Adriana Lúcia Meireles, Eduardo A. F. Nilson.

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
