## [Decision Letter · Decision Letter 0]

18 Apr 2023

PONE-D-22-29107Avoidable diet-related deaths and cost-of-illness with culturally optimized modifications in diet: the case of Brazil.PLOS ONE

Dear Dr. Eliseu Verly,

Thank you for submitting your manuscript to PLOS ONE. After careful consideration, we feel that it has merit but does not fully meet PLOS ONE’s publication criteria as it currently stands. Therefore, we invite you to submit a revised version of the manuscript that addresses the points raised during the review process.

The authors did a good job on discussion about implications of the study results, however for a manuscript to be published in PLOS-ONE methods must be described in sufficient detail for another researcher to reproduce the study described.

We look forward to receiving your revised manuscript.

Kind regards,

Chulaporn Limwattananon, Ph.D.

Academic Editor

PLOS ONE

Journal Requirements:

2. We noted in your submission details that a portion of your manuscript may have been presented or published elsewhere. [able 1 reproduces some of the relative risks published in Scarborough et al, reference #15.

This is not a dual publication once the table is referenced to the original publication.] Please clarify whether this [conference proceeding or publication] was peer-reviewed and formally published. If this work was previously peer-reviewed and published, in the cover letter please provide the reason that this work does not constitute dual publication and should be included in the current manuscript.

Additional Editor Comments (if provided):

This study looked at optimized diet components that were accounted for food prices and preferences and estimated the number of deaths and economic burden and their cost saving from the health system. The methodology is very crucial for credibility of the results. The estimation of these important results for policy decision requires complex approached. However, the approaches were not well described in the current status of the manuscript, especially the attributable cost. In general, important parameters put into the modelling and their corresponding data sources should be listed in the manuscript. The authors often provided the reference of the particular methods used and this leads to insufficient knowledge that readers could earn.

Page 2, line 9, 10: This requires adding more references. (i.e., There was only reference no. 9.)

9 Several studies have performed diet optimization to design diets attaining nutritional goals

10 while respecting other aspects of the diets, such as food preferences and cost constraints.

Page 5, line 21-22: According to the sentence below, the number of sample, number of food subtypes that were diet risk factors, and observed nutrient components of each age-sex subgroup should be presented.

1591 Food items—115 Food subtypes---26 Food subtypes that were diet risk factors (Table 3)----food quantities (Table 3), energy intake and nutrient component (Table 4)

21 Mean food intakes were obtained for each age-sex group and used as starting points to design

22 optimized diets using linear programming models. Twelve age-sex subgroups were defined: < 30y, 31-

23 39y, 41-49y, 51-59y, 61-69, and >=70, all age groups stratified by sex. The Brazilian Food Composition

24 database (13) was used to obtain nutrient content in both observed and optimized diets. Nutrient

25 composition of foods clustered from food subtypes (e.g. different types of rice into ‘rice’) was obtained

26 as the mean composition of the food subtypes weighted by their frequency of reporting in the NDS.

Page 6, line 18, 19: Spell out for LP and “minimized” should be “minimize”

18 xxxxxxx. The objective function in the LP models was defined to minimized the sum of

19 the absolute difference between each optimized and baseline food quantity, described as follows:

Page 8: Table 1: Please check mmHG should be “mmHg”

Page 8, line 2: Add the abbreviation for PUFAs, and mmHG should be “mmHg”

Page 9: Table 2: In Table 2, please check the unit and consistency with Table 1 and line 7, and the correctness of line 7. For sodium in Table 4, the unit was mg. In addition, the unit per day should be labeled.

7 a percentage in relation to the baseline, in grams for fats and milligrams for cholesterol and sodium.

Page 10, line 25: The calculation of attributable death was based on “outcome-risk pair and age-sex group.” Therefore, the results of these should be shown in the manuscript.

Page 10, line 9-11: How was the Trans Fat Macrosimulation used in this study? Please provide more explanation

9 xxxxxxxxxxx model and the Trans Fat Macrosimulation Model is a comparative risk

10 assessment methodology suitable to estimate the health impact of changes in the behavioral risk factors

11 for non-communicable diseases on population and age-sex specific NCD mortality.

Page 11, line 12-15: Please provide brief explanation of the comparative risk assessment macrosimulation model and its use/parameters to calculate the attributable cost.

12 The attributable costs were estimated through a comparative risk assessment macrosimulation

13 model to estimate the proportion of direct costs to the Brazilian National Health System

14 (hospitalizations) and indirect costs related to premature deaths in each diet scenario, expanded from a

15 costing tool developed for hypertension and related cardiovascular disease (16).

Page 11, line 15-18: “Sugar was never been stated before in the method section about how were they related to fruit or vegetable”

15 xxxxxxxxxxxxxxxxxxxxxxxxxxxxxxxxxxxxxxxxxxxxxxxx. The estimation of the

16 direct and indirect costs related to dietary risks (salt, fats, sugars, fruits, vegetables, and fibers) applied

17 the same PAFs related to each health outcome in each age-sex group for each counterfactual scenario

18 as estimated previously using the PRIME.

Page 12, line 8-9: only the relative risks were varied. Were there other parameters that were also varied for sensitivity analysis.

8 Uncertainty intervals are calculated based on 5,000 Monte Carlo iterations which allows the

9 relative risks to randomly vary according to the distribution described in the literature.

Which software was used for each calculation?

Page 12, line 17-19: 1) Spell out for PUFA; 2) All results of each age sex group pertaining this issue should be presented in the Table so that the contents in these lines can be understood.

17 constraints was necessary. Cholesterol constraint was not attained in 70y+ Females; sodium constraint

18 was not attained in 50-60y Females; PUFA constraints were not attained in 30-40y Males and 60y+

19 Females. In all these cases, the constraints were set to be equal or higher than the baseline mean intakes.

Table 3 and Table 4: The unit per day should be labeled.

Page 15, Table 4: Please provide the abbreviation list for Cho., MUFAs and PUFAs

Page 16, line 1-2: Spell out MUFA and PUFA. However, MUFA was not stated in Table 1.

1 fibers, followed by the changes in fats (lowering total, saturated and trans fats, and cholesterol, while

2 increasing MUFA and PUFA) and salt.

Page 16, Table 5: According to Table 1, could the results be broken down by the individual diseases?

The format of the references must be conformed with the journal requirement.

The availability of the data: The summary of important parameters should be presented in the manuscript.

Yes - all data are fully available without restriction (The datasets analyzed during the current study are available on the Brazilian Institute of Geography and Statistics (IBGE) website, [https://www.ibge.gov.br/en/statistics/social/population/25610-pof-2017-2018-pof en.html?=&t=o-que-e]. Variable names, description, and contents are in Portuguese. The Excel file used for running the PRIME is available on https://www.euro.who.int/en/health-topics/disease prevention/tobacco/publications/2019/ncdprime-modelling-the-impact-of-national policies-on-noncommunicable-disease-ncd-mortality-using-prime-a-policy-scenario modelling-tool-2019. SAS codes used for diet optimization are available from the corresponding author on reasonable request.)

Reviewers' comments:

Reviewer's Responses to Questions

**Comments to the Author**

1. Is the manuscript technically sound, and do the data support the conclusions?

Reviewer #1: Yes

2. Has the statistical analysis been performed appropriately and rigorously? 

Reviewer #1: Yes

3. Have the authors made all data underlying the findings in their manuscript fully available?

Reviewer #1: Yes

4. Is the manuscript presented in an intelligible fashion and written in standard English?

Reviewer #1: Yes

5. Review Comments to the Author

Reviewer #1: Overall summary

This is an interesting study. The study evaluated the impact of change in dietary risk factors on health outcomes and economic burden related to noncommunicable chronic diseases in Brazil by setting the goal at optimized diet. This study used linear programming model to design healthier diet and introduced both dietary constraints and food acceptability constraints into the model. The data sources were based on the national survey. The paper is generally well structured and provided valuable information however there are some points that should be added more explanation.

Method section:

1. Is the study sample for baseline dietary intakes equal to 20,112 households (from NDS)?

2. The total number of study sample and the total number of food items of each age-sex subgroup should be provided in Supplementary section.

3. The final list of food items in each age-sex subgroup ranged from 112 to 119. The author should explain why the total number of food items shown in Table 3 was 26.

4. In Diet modeling section (line 26), the author should add more explanation about how many food items were put into the linear programming model and how to select the food items that were put into the model.

5. The topic of Dietary constraints and Food acceptability constraints should be the subtopic of Diet modeling.

6. In Dietary constraint section, the author should add more details about how to develop a set of key healthy dietary component. The food items that are considered as the dietary risks for NCD based on PRIME model are selected (as shown in Table 2), isn’t it? In addition, the author should add more explanation about the differences among the five scenarios, whether it is gradually increased the nutritional profile of diet and which scenario that were assumed to provide the maximum benefit on health outcome.

7. Table 1 should be moved to Supplementary section.

8. In Attributable death section, the author should add more details about how TFA macrosimulation model were used for calculation attributable death. Why was it used as complementary to PRIME model?

9. In Attributable cost section, comparative risk assessment macrosimulation model refer to TFA macrosimulation model, isn’t it?

10. In Attributable cost section, line 16, why was sugar also considered?

11. In Attributable cost section, line 22, the author should provide how much the data from the Brazilian National Health System represent hospitalizations of NCD patients in Brazil.

12. The author should add the reference for discount rate (2% per year).

Result section:

13. In Table 5 and 6, the author should show the data by NCD diseases such as preventable mortality and hospitalization among patients with CHD, stroke, cancer.

14. Unit of preventable deaths should be clarified in Table 5 (per day or per year). In addition, Page 16, line 13-15, the author should add more explanation about how to calculate the number of deaths would be avoided, for example, 0.023 refer to….

15. In Table 6, trend of saving costs in hospitalizations should be similar to trends of preventable death (Table 5) and saving costs of productivity loss (Table 7). Why saving costs in hospitalization in Scenario #4 was equal to zero?

Other issues required minor change:

o In Table 1, head row in column #1, “Risk factor” should be replaced by “Dietary risk factor”.

o In Table 1, the values in the fourth column for saturated fat, PUFAs, Dietary cholesterol, salt is regression parameter.

o In Table 1, unit of change of dietary cholesterol should be 1 mg/day.

o In Table 2, the author should add the quantities of each dietary item as per day or per…

o In Table 3, order of food items was based on type of food, isn’t it?

o In Table 3, why FV (fruits and vegetables) were grouped?

o In Table 4, there were two type of data, the first part presented total Kcal (nutrient intake) and %Kcal from each nutrient and the second part presented mg or g (nutrient content). The author should add the topic such as nutrient intake and nutrient content in each part. In addition, why sum of %Kcal from each nutrient is approximately 128.66%, which is greater than 100%?

References:

o Introduction section, please add reference in Line 23-24 page 3, line 10 page 4.

o The author should format the references according to the guideline of the journal.

Typing error:

o In Discussion section, page 18, line 5 (0,02/day), line 8 (1,03% and 4,63%), the comma used in the numbers should be revised.

o In Discussion section, page 21, the word “under-reporting” and underreporting” should be consistent.

6. PLOS authors have the option to publish the peer review history of their article (what does this mean?). If published, this will include your full peer review and any attached files.

Reviewer #1: No

---

## [Author Response · Author response to Decision Letter 0]

4 Jun 2023

This study looked at optimized diet components that were accounted for food prices and preferences and estimated the number of deaths and economic burden and their cost saving from the health system. The methodology is very crucial for credibility of the results. The estimation of these important results for policy decision requires complex approached. However, the approaches were not well described in the current status of the manuscript, especially the attributable cost. In general, important parameters put into the modelling and their corresponding data sources should be listed in the manuscript. The authors often provided the reference of the particular methods used and this leads to insufficient knowledge that readers could earn.

Answer: Thank you very much for your careful reading and for the several observations and suggestions for this manuscript. We also i) reviewed the analysis thoroughly and updated some numbers throughout the text; ii) replaced PAF (population attributable fraction) with PIF (potential impact fraction) because it is the appropriate term in this context. Much of the information required by the reviewers was provided as Supplementary Information. We hope this new version brings a clearer description of the procedures and results.

Page 2, line 9, 10: This requires adding more references. (i.e., There was only reference no. 9.)

Answer: Done.

Page 5, line 21-22: According to the sentence below, the number of sample, number of food subtypes that were diet risk factors, and observed nutrient components of each age-sex subgroup should be presented.

Ok – Done. This information was included in Supplementary Information.

Page 6, line 18, 19: Spell out for LP and “minimized” should be “minimize”

Answer: Done.

Page 8: Table 1: Please check mmHG should be “mmHg”

Answer: Done.

Page 8, line 2: Add the abbreviation for PUFAs, and mmHG should be “mmHg”

Answer: Done.

Page 9: Table 2: In Table 2, please check the unit and consistency with Table 1 and line 7, and the correctness of line 7. For sodium in Table 4, the unit was mg. In addition, the unit per day should be labeled.

Answer: The unity for sodium is mg. In table 2 the scenarios 4 and 5 represent change in % in relation to the baseline intake. The same for PUFA, saturated fat, and cholesterol. That is why they are not presenting in mg or %kcal. 

Page 10, line 25: The calculation of attributable death was based on “outcome-risk pair and age-sex group.” Therefore, the results of these should be shown in the manuscript. 

Answer: We agree that these results should also be presented. We included in Supplementary Information.

Page 10, line 9-11: How was the Trans Fat Macrosimulation used in this study? Please provide more explanation

Thanks for this observation. It was a mistake when describing the procedure. We corrected it throughout the text. 

Page 11, line 12-15: Please provide brief explanation of the comparative risk assessment macrosimulation model and its use/parameters to calculate the attributable cost. 

Answer: Done. To estimate the attributable cost, we applied the same PIF (potential impact fraction) applied to estimate the number of deaths. We made it clearer in the manuscript. 

Page 11, line 15-18: “Sugar was never been stated before in the method section about how were they related to fruit or vegetable”

Thanks for this observation. It was a mistake when describing the procedure. We corrected it throughout the text. 

Page 12, line 8-9: only the relative risks were varied. Were there other parameters that were also varied for sensitivity analysis.

Answer: It considered the variation in the RR and the dietary inputs. We made it clearer in the manuscript. 

Which software was used for each calculation?

Answer: The SAS software was used to run the linear programming models and descriptive analysis. The Microsoft Excel was used to run the PRIME spreadsheet. This information was described in the main text.

Page 12, line 17-19: 1) Spell out for PUFA; 2) All results of each age sex group pertaining this issue should be presented in the Table so that the contents in these lines can be understood.

Answer: Done.

Table 3 and Table 4: The unit per day should be labeled.

Answer: Done.

Page 15, Table 4: Please provide the abbreviation list for Cho., MUFAs and PUFAs

Answer: Done.

Page 16, line 1-2: Spell out MUFA and PUFA. However, MUFA was not stated in Table 1.

Answer: MUFA was not included in the optimization model. However, MUFA content varied across the scenarios due to changes in other fat compounds as indicated in Table 2, i.e., due to food and fats replacement to meet the dietary constraints. “Spontaneous” changes in MUFA content across the scenarios were accounted when computing the attributable fraction. 

Page 16, Table 5: According to Table 1, could the results be broken down by the individual diseases?

Answer: Done. Presented in Supplementary Information

Reviewers' comments:

Overall summary

This is an interesting study. The study evaluated the impact of change in dietary risk factors on health outcomes and economic burden related to noncommunicable chronic diseases in Brazil by setting the goal at optimized diet. This study used linear programming model to design healthier diet and introduced both dietary constraints and food acceptability constraints into the model. The data sources were based on the national survey. The paper is generally well structured and provided valuable information however there are some points that should be added more explanation. 

Answer: Thank you very much for your careful reading and for the several observations and suggestions for this manuscript. We also i) reviewed the analysis thoroughly and updated some numbers throughout the text; ii) replaced PAF (population attributable fraction) with PIF (potential impact fraction) because it is the appropriate term in this context. Much of the information required by the reviewers was provided as Supplementary Information. We hope this new version brings a clearer description of the procedures and results.

Method section:

1. Is the study sample for baseline dietary intakes equal to 20,112 households (from NDS)?

Answer: Yes, the total of households visited for dietary intake collection was 20,112. However, for this study, dietary information was used only for individuals 25y or older (n=37,687). We made it clearer in the text. 

2. The total number of study sample and the total number of food items of each age-sex subgroup should be provided in Supplementary section.

Answer: Done.

3. The final list of food items in each age-sex subgroup ranged from 112 to 119. The author should explain why the total number of food items shown in Table 3 was 26.

Answer: In table 3, food list was aggregated in main foods or food groups. Also, we revised the analysis thoroughly and corrected the number of foods, which now varies from 83 to 85.

4. In Diet modeling section (line 26), the author should add more explanation about how many food items were put into the linear programming model and how to select the food items that were put into the model. 

Answer: All the reported foods (after initial aggregation, as described in line 21, pg 5) were included into the model with exception to coffee, tea, and alcoholic beverages. The number of foods varied according to the age-sex group because not every food was reported by individuals in all age-sex groups. The number of foods put into the model varied from 83 to 85. 

5. The topic of Dietary constraints and Food acceptability constraints should be the subtopic of Diet modeling.

Answer: Done.

6. In Dietary constraint section, the author should add more details about how to develop a set of key healthy dietary component. The food items that are considered as the dietary risks for NCD based on PRIME model are selected (as shown in Table 2), isn’t it? In addition, the author should add more explanation about the differences among the five scenarios, whether it is gradually increased the nutritional profile of diet and which scenario that were assumed to provide the maximum benefit on health outcome.

Answer: Thanks for this comment. Yes, we defined the scenarios based on the dietary risks of PRIME model. The main rationale for the changes across the scenarios was to evaluate moderate and gradual modifications in the current dietary intakes. There is no evidence on what is in fact moderate, feasible or acceptable in a population, thus we increased or decrease the amount of each dietary risk in small to moderate amounts that, at our best judgment, are potentially achievable at the population level. More drastic changes would provide more benefits; however, they would cost more, which is, itself, a barrier to being adopted. 

7. Table 1 should be moved to Supplementary section.

Answer: Thanks for this suggestion. However, we opted to keep the table in the main text because it summarizes the dietary risks and outcomes, which, to our judgment, are essential information for the readers.

8. In Attributable death section, the author should add more details about how TFA macrosimulation model were used for calculation attributable death. Why was it used as complementary to PRIME model?

Thanks for this observation. It was a mistake when describing the procedure. We corrected it throughout the text. 

9. In Attributable cost section, comparative risk assessment macrosimulation model refer to TFA macrosimulation model, isn’t it?

Answer: It was a mistake when describing the procedure. We corrected it throughout the text.

10. In Attributable cost section, line 16, why was sugar also considered?

Answer: Thanks for this observation. It was a mistake when describing the procedure. We corrected it throughout the text. 

11. In Attributable cost section, line 22, the author should provide how much the data from the Brazilian National Health System represent hospitalizations of NCD patients in Brazil.

Answer: Done. However, there is no data on how Much it represents by cause of deaths. The available data is regarding to the total of hospitalization, which is about 70% provided by the National Health System. 

12. The author should add the reference for discount rate (2% per year). 

Answer: Done.

Result section:

13. In Table 5 and 6, the author should show the data by NCD diseases such as preventable mortality and hospitalization among patients with CHD, stroke, cancer.

Answer: We agree that these results should also be shown. We included in Supplementary Information.

14. Unit of preventable deaths should be clarified in Table 5 (per day or per year). In addition, Page 16, line 13-15, the author should add more explanation about how to calculate the number of deaths would be avoided, for example, 0.023 refer to….

Answer: Done. We clarified this sentence in the text. 

15. In Table 6, trend of saving costs in hospitalizations should be similar to trends of preventable death (Table 5) and saving costs of productivity loss (Table 7). Why saving costs in hospitalization in Scenario #4 was equal to zero? 

Answer: Thanks for noticing this mistake. We corrected the table. 

Other issues required minor change:

o In Table 1, head row in column #1, “Risk factor” should be replaced by “Dietary risk factor”.

Answer: Done.

o In Table 1, the values in the fourth column for saturated fat, PUFAs, Dietary cholesterol, salt is regression parameter.

Answer: Thanks. Done.

o In Table 1, unit of change of dietary cholesterol should be 1 mg/day.

Answer: The “unit of change” column refers to the changes in dietary intakes (gram or milligram/day) and the “outcome” column, in this case, refers to the total serum cholesterol, in nmol/L.

o In Table 2, the author should add the quantities of each dietary item as per day or per…

Answer: Done.

o In Table 3, order of food items was based on type of food, isn’t it?

Answer: Yes.

o In Table 3, why FV (fruits and vegetables) were grouped?

Answer: We opted to aggregate them because FV is a marker of healthy diet and there is general intake recommendation for this group. We also presented them apart. 

o In Table 4, there were two type of data, the first part presented total Kcal (nutrient intake) and %Kcal from each nutrient and the second part presented mg or g (nutrient content). The author should add the topic such as nutrient intake and nutrient content in each part. In addition, why sum of %Kcal from each nutrient is approximately 128.66%, which is greater than 100%?

Answer: A very little variation around 100% is expected due to methodological issues. In this case, summing up the macronutrients (carbohydrates, total fats, and protein) results in (54.8+28.8+19.2) = 102.8%. The percentage from saturated fat, MUFA, PUFA, and trans-fats should not be summed up with the other macronutrients because they are types of fat, therefore accounted in the energy contribution of fats.

References:

o Introduction section, please add reference in Line 23-24 page 3, line 10 page 4.

Answer: Done. A reference in line 10 page 4 was placed at the end of the next sentence. 

o The author should format the references according to the guideline of the journal.

Answer: Done.

Typing error:

o In Discussion section, page 18, line 5 (0,02/day), line 8 (1,03% and 4,63%), the comma used in the numbers should be revised.

Answer: Thanks. Done.

o In Discussion section, page 21, the word “under-reporting” and underreporting” should be consistent.

Answer: Thanks. Done.

---

## [Decision Letter · Decision Letter 1]

25 Jun 2023

PONE-D-22-29107R1Avoidable diet-related deaths and cost-of-illness with culturally optimized modifications in diet: the case of Brazil.PLOS ONE

Dear Dr. Verly-Jr,

Thank you for submitting your manuscript to PLOS ONE. After careful consideration, we feel that it has merit but does not fully meet PLOS ONE’s publication criteria as it currently stands. Therefore, we invite you to submit a revised version of the manuscript that addresses the points raised during the review process.

ACADEMIC EDITOR: Please find comments below

We look forward to receiving your revised manuscript.

Kind regards,

Charles Odilichukwu R. Okpala

Academic Editor

PLOS ONE

Journal Requirements:

Additional Editor Comments:

Please authors, kindly address the minor concerns. Please also re-check the grammar of text throughout, to sharpen it further ok.

Reviewers' comments:

Reviewer's Responses to Questions

**Comments to the Author**

1. If the authors have adequately addressed your comments raised in a previous round of review and you feel that this manuscript is now acceptable for publication, you may indicate that here to bypass the “Comments to the Author” section, enter your conflict of interest statement in the “Confidential to Editor” section, and submit your "Accept" recommendation.

Reviewer #1: All comments have been addressed

Reviewer #2: All comments have been addressed

2. Is the manuscript technically sound, and do the data support the conclusions?

Reviewer #1: Yes

Reviewer #2: Yes

3. Has the statistical analysis been performed appropriately and rigorously? 

Reviewer #1: Yes

Reviewer #2: Yes

4. Have the authors made all data underlying the findings in their manuscript fully available?

Reviewer #1: Yes

Reviewer #2: Yes

5. Is the manuscript presented in an intelligible fashion and written in standard English?

Reviewer #1: Yes

Reviewer #2: Yes

6. Review Comments to the Author

Reviewer #1: Thank you for the opportunity to review the revised manuscript. The author has thoroughly revised the manuscript in accordance with the comments from the reviewers in the first round and also added relevant information in supplementary material.

Minor concern is about typing error.

Example:

Page 5: Line 20: From this list, the items were aggregated if they were different type, different cooking method, or different meat cut of the same food or (e.g., different types of orange into 21 “orange”, different beef cuts into “beef”, etc.).

Table 4: Charbohydrates (%Kcal)

Reviewer #2: The sub-heading "Comparison with other studies" is not necessary. Paragraph delineation introducing comparison is better

7. PLOS authors have the option to publish the peer review history of their article (what does this mean?). If published, this will include your full peer review and any attached files.

Reviewer #1: No

Reviewer #2: No

---

## [Author Response · Author response to Decision Letter 1]

26 Jun 2023

Additional Editor Comments:

Please authors, kindly address the minor concerns. Please also re-check the grammar of text throughout, to sharpen it further ok.

Answer: thank you very much. We carefully revised the manuscript thoroughly (grammar and other unobserved typos). We also identified and corrected few inconsistencies between the results in the text and tables. 

Thank you for the opportunity to review the revised manuscript. The author has thoroughly revised the manuscript in accordance with the comments from the reviewers in the first round and also added relevant information in supplementary material. 

Minor concern is about typing error. 

Example: 

Page 5: Line 20: From this list, the items were aggregated if they were different type, different cooking method, or different meat cut of the same food or (e.g., different types of orange into 21 “orange”, different beef cuts into “beef”, etc.).

Table 4: Charbohydrates (%Kcal)

Answer: thank you very much for observing these mistakes. We carefully revised the manuscript thoroughly (grammar and other unobserved typos).

---

## [Editor Report · Decision Letter 2]

29 Jun 2023

Avoidable diet-related deaths and cost-of-illness with culturally optimized modifications in diet: the case of Brazil.

PONE-D-22-29107R2

Dear Dr. Verly-Jr,

We’re pleased to inform you that your manuscript has been judged scientifically suitable for publication and will be formally accepted for publication once it meets all outstanding technical requirements.

Kind regards,

Charles Odilichukwu R. Okpala

Academic Editor

PLOS ONE

Additional Editor Comments (optional):

Thank you for revising your work, and addressing all concerns raised by the reviewers

Acceptable for publication
---

## [Editor Report · Acceptance letter]

3 Jul 2023

PONE-D-22-29107R2 

Avoidable diet-related deaths and cost-of-illness with culturally optimized modifications in diet: the case of Brazil. 

Dear Dr. Verly-Jr:

I'm pleased to inform you that your manuscript has been deemed suitable for publication in PLOS ONE. Congratulations! Your manuscript is now with our production department. 

Kind regards, 

on behalf of

Dr. Charles Odilichukwu R. Okpala 

Academic Editor

PLOS ONE